# Expanding the drug discovery space with predicted metabolite–target interactions

Andrea Nuzzo [1✉], Somdutta Saha[1,3], Ellen Berg [2], Channa Jayawickreme[1], Joel Tocker[1] & James R. Brown [1,4✉]

Metabolites produced in the human gut are known modulators of host immunity. However, large-scale identification of metabolite–host receptor interactions remains a daunting challenge. Here, we employed computational approaches to identify 983 potential metabolite–target interactions using the Inflammatory Bowel Disease (IBD) cohort dataset of the Human Microbiome Project 2 (HMP2). Using a consensus of multiple machine learning methods, we ranked metabolites based on importance to IBD, followed by virtual ligand-based screening to identify possible human targets and adding evidence from compound assay, differential gene expression, pathway enrichment, and genome-wide association studies. We confirmed known metabolite–target pairs such as nicotinic acid–GPR109a or linoleoyl ethanolamide–GPR119 and inferred interactions of interest including oleanolic acid–GABRG2 and alpha-CEHC–THRB. Eleven metabolites were tested for bioactivity in vitro using human primary cell-types. By expanding the universe of possible microbial metabolite–host protein interactions, we provide multiple drug targets for potential immune-therapies.

[1] GlaxoSmithKline Pharma R&D, 1250 S. Collegeville Rd, Collegeville, PA 19426-0989, USA. [2] Eurofins Discovery, 111 Anza Boulevard, Burlingame, CA 94010, USA. [3] Present address: EMD Serono Research & Development Institute, Inc. 45A Middlesex Turnpike, Billerica, MA 01821, USA. [4] Present address: Kaleido Biosciences, Inc. 65 Hayden Avenue, Lexington, MA 02421, USA. ✉email: andrea.8.nuzzo@gsk.com; jim.brown@kaleido.com

Endogenous metabolites produced in the gastro-intestinal tract (GIT) by microbial and human metabolic processes have a significant role in modulating host immune responses[1]. Therefore, targeting the interspecies cross-talk between microbial metabolites and human host receptors holds a recognized therapeutic potential[2]. Disentangling these interactions in order to retrieve meaningful information remains highly challenging[3]. Observational studies of microbial metabolite abundances in human disease-related cohorts can suggest general associations with disease etiology but lack the granularity to identify specific metabolite–host receptor pairings[4] or causal relationships[5]. Conversely, mechanistic studies that have either focused on a few selected metabolites screened against specific receptors[6], or adopted system biology modeling[7] are limited by our current knowledge of metabolic pathways and are difficult to scale to identify hundreds or thousands of interactions with druggable potential.

Growing in vitro receptor–ligand assay databases have greatly increased the hypothesis space for drug discovery[8] while giving meaningful mechanistic information regarding metabolite–protein interactions. Here we present the results of a large-scale computational analyses, using bioinformatic and chemoinformatic approaches, to identify multiple and specific interactions between endogenous metabolites and human host proteins. We utilized a multi-omics dataset from an Inflammatory Bowel Disease (IBD) cohort[9] published by the Human Microbiome Project 2 consortium (HMP2)[10]. IBD includes Ulcerative Colitis (UC) and Crohn's Disease (CD), whose etiology heavily depends on the interplay between the GIT microbiome and immune system[11]. We then used virtual ligand-based screening to predict the activity of the original query metabolites on multiple targets[12] based upon historical assay databases containing interaction data between similar molecules and specific targets. We believe that these predicted interactions will further our understanding of host–microbiome interactions as well as assist in drug discovery for IBD and other diseases.

## Results

### Ranking metabolites for relative importance in disease states.

The HMP2 consortium IBD cohort sub-study[9] involved intensive multi-omics characterization of patients with CD or UC and non-IBD control subjects. We focused on the metabolomics data collected from stool over the course of one year (specifically, the pre-processed metabolomics abundance tables) and bulk transcriptomics data obtained from biopsies of different sections of the gut at the beginning of the study (specifically, the pre-processed transcriptomic count tables) (see Methods). Patients with less than 3 samples per datatype or with only one sampling point were excluded, to the final sample size described in Table 1. Each sample included data on 548 metabolites (matched against the Human Metabolome Database [HMDB]), and 43870 transcripts (aligned to Genome Reference Consortium Human Build 37 [GRCh37]).

Metabolomic samples did not cluster effectively by disease state in embedded projections (Fig. 1a). To better define metabolite relevance for CD or UC etiology we utilized an ensemble method that combined results from multiple analytical methods (specifically, power estimation and both feature importance and SHAP values, each from two selected machine learning methods) combined into a single consensus score, normalized between 0 and 1, where 1 represents the most significant metabolite across all methods (Supplementary Figs. 1 and 2; Supplementary Data 1). Using power estimation alone, approximately 73% of the 548 metabolites were differentially abundant (log-fold change with $q$ value ≤ 0.05) between non-IBD controls and either CD or UC patients (Fig. 1b). Metabolites in the top quartile of the consensus scoring, 29% of which overlapped with significantly differentially abundant metabolites detected in the original HMP2-IBD study[9], were considered for downstream analysis ($n = 192$) (Fig. 1c). We also annotated the selected 192 metabolites from HMDB, but no statistical difference was found in the consensus scoring across metabolite ontology (Supplementary Fig. 3).

In agreement with previous studies, CD and UC patients had a significant (i.e., $q$ value < 0.05) depletion of short chain fatty acids like butyrate ($\log_{10}fc = -0.15$[CD]; $-0.06$[UC]) and valerate/isovalerate ($\log_{10}fc = -0.29$[CD]; $-0.48$[UC]). Significant enrichments included several acylcarnitines, arachidonate ($\log_{10}fc = 0.59$[CD]; $0.60$[UC]), taurocholate ($\log_{10}fc = 0.16$[CD]; $0.04$[UC]), with a correspondent depletion of lithocolate ($\log_{10}fc = -0.30$[CD]; $-0.58$[UC]) and deoxycholate ($\log_{10}fc = -0.53$[CD]; $-0.53$[UC]). Among the highest-ranking metabolites in CD and UC, we also detected a substantial and significant enrichment in nicotinuric acid, ($\log_{10}fc = 0.43$[CD]; $1.16$[UC]), C18:1 carnitine ($\log_{10}fc = 0.54$[CD]; $0.30$[UC]) and several triacylglycerols, while porphobilinogen ($\log_{10}fc = -0.26$[CD]; $-0.70$[UC]), tetradecanedioic acid ($\log_{10}fc = -0.76$[CD]; $-0.86$[UC]) and nicotinic acid ($\log_{10}fc = -0.49$[CD]; $-0.99$[UC]) were notably depleted.

### Connecting metabolites as ligands to specific human proteins.

To connect metabolites with perspective receptors, we used their chemical structure as query to find structurally similar natural or synthetic compounds with functional assay data in the ChEMBL database. The original metabolites were mapped to ChEMBL compounds by similarity searches using canonical chemical fingerprints which reduced the total space of metabolites and perspective targets to 13,082 unique pairs. We further filtered for compounds having high similarity scores with the top-ranking metabolites (i.e., Tanimoto similarity ≥ 0.85 or Tversky$_{\alpha=0.05}$ similarity ≥ 0.95) and, for those, only binding proteins with perspective high affinity (i.e., either pIC$_{50}$ or pEC$_{50}$ values, hereafter referred to as pxC$_{50}$ ≥ 5.5) were retained. Highly pleiotropic metabolites (i.e., metabolites with ≥ predicted 20 targets), or pleiotropic targets (i.e., targets with predicted associations to ≥ 20 metabolites) were removed to further refine the hypothesis space.

A total of 135 metabolites were provisionally connected to 80 different perspective targets (Fig. 2a; Supplementary Data 2). Those metabolites fell mostly into the lipid-like and organic acid

**Table 1 Overview of the samples and patient dataset used in this study (after filtering from the original study by Lloyd-price et al[9].).**

| Diagnosis | Participants by sampling | | Total samples over time | |
| --- | --- | --- | --- | --- |
| | Metabolomics | Host transcriptomics | Metabolomics | Host transcriptomics |
| CD | 43 | 50 | 127 | 265 |
| UC | 25 | 30 | 74 | 146 |
| nonIBD | 22 | 26 | 51 | 135 |
| Total | 90 | 106 | 252 | 546 |

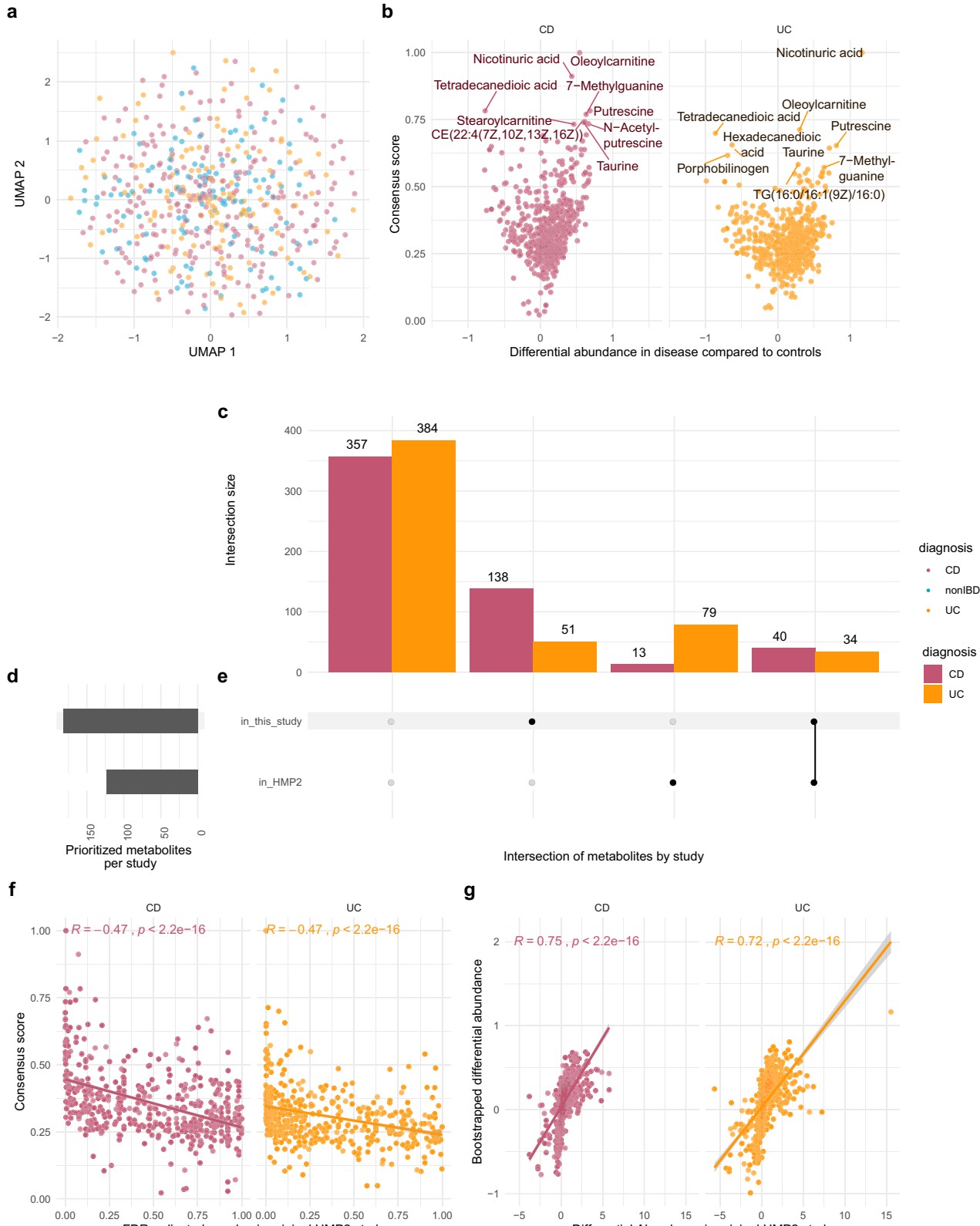

**Fig. 1 Metabolomics results and comparisons to the original HMP2 IBD study[9] (HMP2). a** UMAP analysis of the metabolomics sample distribution by Crohn's disease (CD) and Ulcerative Colitis (UC) patients and controls (nonIBD). **b** Volcano plot showing the differential abundance of each metabolite per disease state against the consensus scoring of each state. **c** Number of metabolites considered relevant in HMP2 and current study per disease state, subdivided into overlapping and non-overlapping subsets. **d** Total number of metabolites selected relevant in each study. **e** Intersection matrix between metabolites selected each study. **f** Correlation plot between the bootstrapped power estimation method used to determine metabolite differential abundance between CD and UC patients results. **g** Correlation between the consensus scoring used in this study and HMP2 FDR-adjusted *p* values for each metabolite (refer to Table 1 for samples composition).

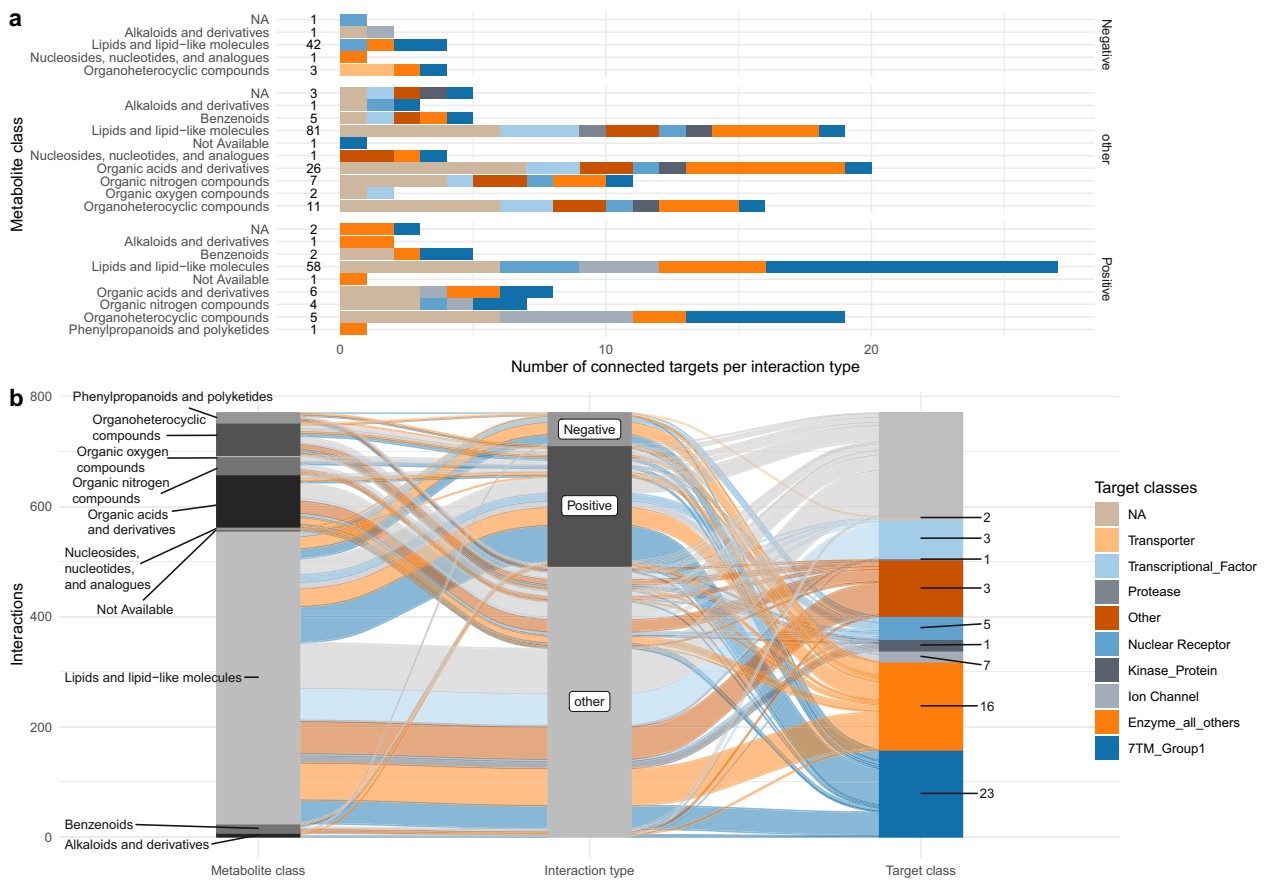

**Fig. 2 Overview of the connected metabolites with highest ranking scores and the perspective targets. a** Distribution of target drug classes per each metabolite class (numbers represent unique metabolites per metabolite class). **b** Alluvial plot describing the distribution of connections between metabolite classes, modulation type and drug target classes (numbers represent unique targets per drug target class).

macro-categories with expected modulatory activity against 9 macro-categories of drug targets (Fig. 2b). For example, 7-methylguanine, is structurally similar (Tversky$_{\alpha=0.05}$ similarity = 0.96) to 8-aminoguanine (CHEMBL8040) which is an inhibitor (pxC$_{50}$ = 5.8–6.2) of purine nucleoside phosphorylase (PNP). Among the lipid-like metabolites, heptanoic acid was connected through azelaic acid (CHEMBL1238, Tversky$_{\alpha=0.05}$ similarity = 0.955) to the nuclear factor kappa-light-chain-enhancer of activated B cells (NF-kB) and peroxisome proliferator-activated receptors alpha (PPARA) with high inhibitory activity (pxC$_{50}$ = 6.0 and 8.6). Hydrocinnamic acid, depleted in both UC and CD patients was connected to cytochrome P450 p1a2 (CYP1A2) (Tanimoto similarity = 0.89) via a strong analog inhibitor (pxC$_{50}$ = 8.3). Nicotinic acid (underrepresented in CD and UC patients) was connected to its known receptor, the hydroxycarboxylic acid receptor 2 (HCAR2 or GPR109a), whereas the product of its degradation, nicotinuric acid, was overrepresented and connected to Lamin A/C protein (LMNA) through a binder analog (Tversky$_{\alpha=0.05}$ similarity = 0.97, pxC$_{50}$=7.65) with unknown directionality. Alpha-carboxyethyl hydroxychroman (alpha-CEHC) was also connected to LMNA but also to the thyroid hormone receptor beta (THRB). However, since the modulatory action of the analog compound is unknown and alpha-CEHC is depleted in UC but enriched in CD, we were unable to infer directionality of the interaction. Oleanolic acid was connected through other plant terpenoids (urolic and moronic acid) and bacterial-specific sphingolipids (i.e., CHEMBL1334750) to several targets of interest for pharmaceutical purposes such as GPBAR1 (G Protein-Coupled Bile Acid Receptor 1) and PTPN7 (Protein tyrosine phosphatase non-receptor type 7).

**Connecting gene expression and metabolite abundance**. We then considered differential expressed genes (DEGs) comparing non-IBD against CD and UC states respectively, accounting for the heavy impact of the biopsy location variable (Fig. 3a). A total of 2170 DEGs occurred overall, of which 820 DEGs were shared by both CD and UC (Fig. 3b). Pathway enrichment analysis determined a high representation of immune inflammation-related pathways (i.e., Cytokine Signaling, NRF2 non−canonical NF − kB pathway, Interleukin 3, 14 and 17 signaling) (Supplementary Fig. 4; Supplementary Data 3).

Starting from DEGs, we proceeded to parse connections with differentially abundant metabolites using the ChEMBL database, by inverting the workflow described above. After parsing all possible modulators among for DEGs, top-ranking metabolites were considered modulators if having any similar analog with functional activity against the candidate gene represented by the transcript, resulting in a total of 45 prospectively druggable targets.

Several metabolites underrepresented in IBD were classified as tentative negative modulators of upregulated targets. For example, receptors of the CXC ligand 8 (CXCL8 or IL8) chemokine family, CXCR1 and CXCR2, were overexpressed while their known negative modulator compounds, ibuprofen (pxC$_{50}$ = 7.0) and its HMP-2 derivative, 2-hydroxibuprofen, (Supplementary Data 3), were under-represented in IBD patients although below the consensus scoring threshold (Supplementary Data 1).

Another example is HCAR2 (GPR109a) which was upregulated in CD and UC patient biopsies (log$_2$fc = 6.15 [CD] and 4.51 [UC]) while its competing modulators, nicotinic acid and trigonelline were depleted and enriched, respectively, in IBD patients' stool (Fig. 3c; Supplementary Data 2).

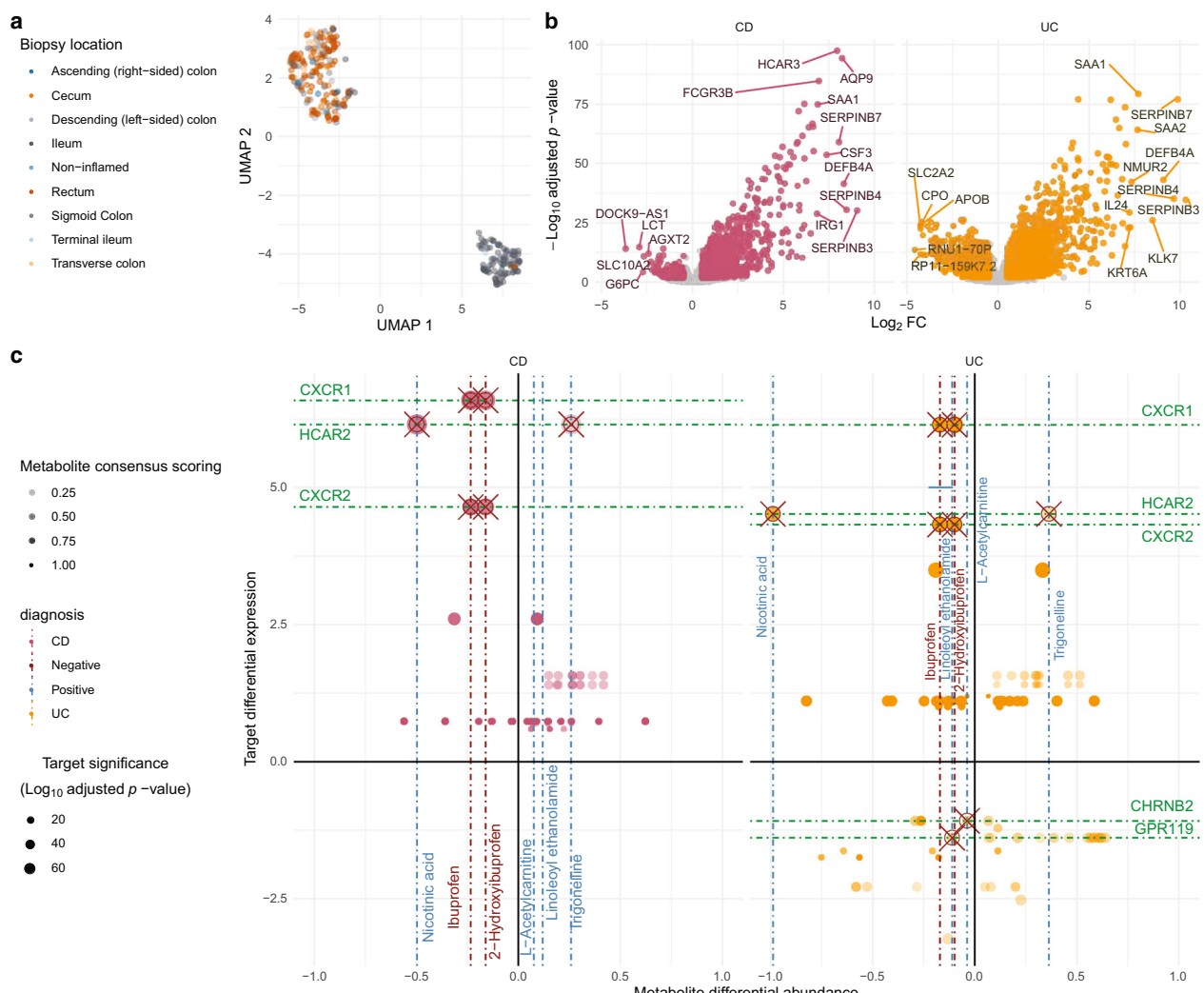

**Fig. 3 Overview of the transcriptomics analysis results. a** UMAP analysis of the transcriptomic samples by biopsy location. **b** Volcano plots representing target differential expressions in Crohn's disease (CD) and Ulcerative Colitis (UC) states by FDR-adjusted $p$ value. **c** Cross-plot showing possible interactions of interest between targets with expression on the vertical axis and perspective modulator metabolite differential abundances on the horizontal axis, per each disease type (refer to Table 1 for samples composition).

We also built connections based on co-directionality between metabolite depletion and corresponding downregulation of perspective targets. For example, GPR119 was downregulated in the UC cohort (log$_2$fc = −1.39) and its known activator, linoleoyl ethanolamide (pxC50 = 5.66) was significantly depleted in UC patients (Supplementary Data 1). Neuronal acetylcholine receptor subunit beta-2 (CHRNB2) was downregulated in UC patients (log$_2$fc = −1.08) where L-acetylcarnitine and cotinine were depleted, both analog to acetylcholine (Tversky$_{\alpha=0.05}$ similarity =0.95 and 0.99), a strong inhibitor of CHRNB2 (pxC$_{50}$ = 8.8). Nitric oxide synthase 2 (NOS2), upregulated in IBD patients (log$_2$fc = 2.60[CD]; 3.49[UC]), was connected to the depletion of negative modulators such as L-arginine (pxC$_{50}$ = 6.52, log$_{10}$fc = −0.31[CD]; −0.19[UC]).

**Assigning candidate metabolites to targets with genetic evidence.** We retrieved 808 genes with genetic association to IBD from the GWAS catalog[13] and an extensive published review of IBD pathways[14]. These genes were intersected with target-compound assay and HMP2 datasets which resulted in 464 potential unique pairings of candidate genetic targets with metabolite modulators (Supplementary Data 4), 13 of which have

metabolites with known modulation mechanisms (Fig. 4a–d; Supplementary Data 5).

CXCR1 and CXCR2 were mapped to a regulatory variant (rs11676348-T) statistically associated with an increase risk to UC[15] (Supplementary Fig. 5), and in our study were mapped to an inhibitor, ibuprofen. An intronic variant statistically associated to inflammatory skin disease (rs4795067)[16] is mapped to NOS2, which is also part of enriched nitric oxide and microbe-sensing pathways, both involved in IBD phenotypes (Fig. 4a); we connected NOS2 with the scarcity of arginine, an inhibitor. Other metabolite–target pairings, although not differentially expressed in the HMP2 dataset, had interesting genetic and metabolomics associations. For instance, an intergenic variant (rs79243092-C) mapped to gamma-aminobutyrate receptor subunit 2 (GABRG2) is linked to an increase in macrophage inflammatory protein 1b in Europeans[17]. In our study we linked GABRG2 to several conjugated bile acids and corticosteroids depleted in IBD, including oleanolic acid, through ganaxolone (CHEMBL1568698) and allopregnanolone (CHEMBL207538), two activator compounds (Fig. 4b). Finally, variant rs56330463-C mapped to the serotonin receptor (HTR4) is associated with increase in monocytes, an inflammatory phenotype[18]. Notably, serotonin was depleted in UC an CD patients while its precursor,

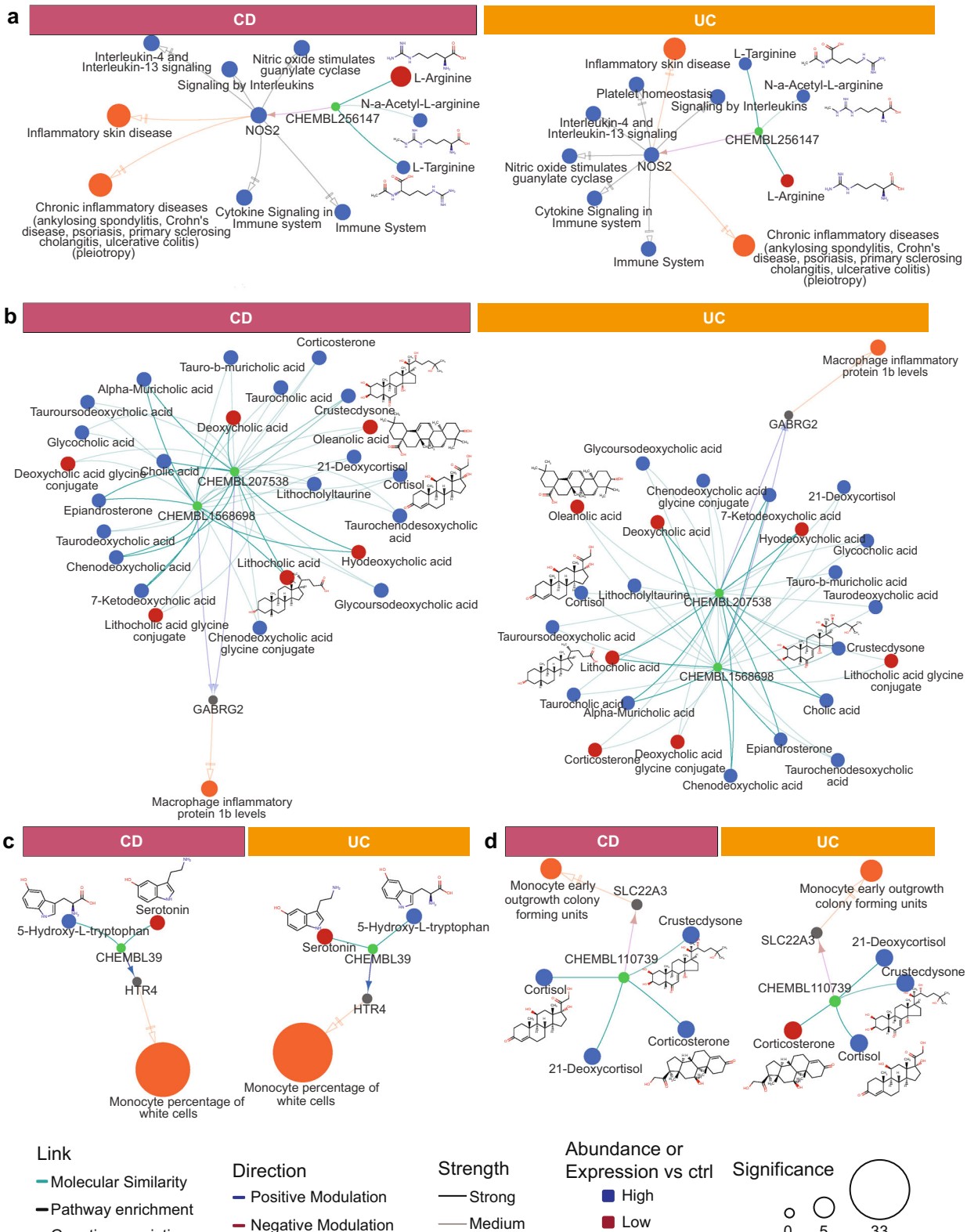

**Fig. 4 Overview of identified for putative target proteins.** Metabolites were connected through similar ChEMBL compounds where similarity is classified as medium (0.8 < Tanimoto score < 0.9) or strong (Tanimoto score > 0.9) for (**a**) CXCR1/2 and NOS2, (**b**) HTR4, (**c**) GABRG2 and (**d**) SLC22A3. Direction and affinity of analog binding to target was parsed from ChEMBL assay databases and represented as medium (5.5 < pxC$_{50}$ < 7.0) and strong (pxC$_{50}$ > 7.0). Direction (i.e., up- or down-) of differential expression for targets or differential abundance for metabolites are represented by colors. Target significance (log $q$ value) is based on differential expression of the gene or GWAS association. Complete results for targets with genetic evidence are shown in Supplementary Fig. 5.

H-hydroxy-L-tryptophan (5-HTP) was enriched (Fig. 4c). Finally, intronic variant rs402219-G, inside the solute carrier family 22 member 3 gene (SLC22A3) is genetically associated to another monocyte-related inflammatory marker;[19,20] SLC22A3 is antagonized by corticosterone (CHEBL110739) and several corticosterone-similar metabolites were enriched in disease states (Fig. 4d). In summary, our collective analyses identified thousands of unique of metabolite–targets pairs, 983 of which having known direction of modulation (Supplementary Data 6, unfiltered results in Supplementary Data 7).

**Biological effects of specific metabolites from complex in vitro cellular assays**. To evaluate the biological activity of predicted receptor–metabolites pairings, we profiled eleven candidate metabolites across a standardized panel of disease-relevant human primary cell-based phenotypic cellular assays, the Bio-MAP® Diversity PLUS panel[21]. The profiled metabolites were 13-cis-retinoic acid, acetylcholine, adenosine, alpha-CEHC, butyrate, histamine, ibuprofen, lithocholic acid, nicotinic acid, oleanolic acid and serotonin (assay results and plots are shown in Supplementary Data 8 and Supplementary Fig. 6, respectively). Different concentrations were tested for each metabolite due to cytotoxicity constraints, but keeping highest concentrations above reported average concentrations measured in human blood (as from the HMDB). None of the compounds showed cytotoxicity effects. However, quantitative differences in the responses of specific cellular systems could be related to the metabolite concentrations which were determined by compound solubility and avoidance of cytotoxicity.

In Fig. 5, we show examples of four metabolites where knowledge-based canonical pathway analyses were used to identify explanatory links between the perspective modulated targets for each metabolite and their respective in vitro assay readouts. Butyrate is a known immunomodulator of GPR41, GPR43 and HCAR2 so we performed in vitro assays for benchmarking purposes. Butyrate showed anti-inflammatory and immunomodulatory related activity (i.e., decreased levels of IL-6, IL-8, IL-10, and TNF-alpha) across multiple cellular assays representing T cell dependent B cell activation (BT), fibroblasts (HDF3CGF) and coronary artery smooth muscle cells (CASM3C) (Fig. 5a), but also had antiproliferative effects on lymphocytes in the BT system.

Nicotinic acid (vitamin B3) is an anti-inflammatory activator of HCAR2. Nicotinic acid was largely inactive at tested concentrations with minor lowering of soluble IL-17A in the BT system (Fig. 5b). Alpha-CEHC was a highly scoring metabolite in our analysis with unclear directionality to disease mechanisms. Alpha-CEHC also had low activity with slight suppression of several inflammatory markers in the HDF3CGF system of dermal fibroblast cells modeling wound healing and fibrosis (Fig. 5c). Finally, we show oleanolic acid which has known anti-inflammatory properties and, based on our analysis, potentially interacts with GABRG2 as its target. Oleanolic acid showed strong anti-inflammatory and immunomodulatory activity across multiple systems including the above-mentioned BT, CASM3C and HDF3CGF systems as well as KF3CT which models Th1 cutaneous inflammation. Interestingly, oleanolic acid was not antiproliferative to lymphocytes in the BT assay, contrary to butyrate. We also mapped oleanolic acid to PTPN7 which can modulate VCAM1 (suppressed in HDF3CGF fibroblast assay) through effects on p38 MAP-kinase (Fig. 5d).

## Discussion

We greatly expand the number of potential protein–metabolite interactions based on a well-characterized IBD multi-omics dataset from HMP2[9] by going beyond conventional associative studies. Machine learning algorithms evaluated the relative importance of each metabolite in the context of the whole metabolomics profile while adoption of a consensus scoring minimized biases across the different methods (Supplementary Fig. 1 and 2). Compared to the HMP2 IBD study, our consensus scoring retrieved more relevant metabolites without recurring to a dysbiosis score, an index derived from the beta-diversity analyses of the metagenomic specimens, which poses issues for reproducibility across cohorts and translatability to treatment purposes[22]. Patterns of enrichments and depletion expected for UC and CD from the literature or the HMP2 study, were confirmed by our method, such as the depletion of short chain fatty acids like butyrate and valerate in CD and UC[23]. Thus, we believe that consensus scoring for each metabolite across the whole metabolomic signature of the disease state better defines the hypothesis space for metabolite–target interactions. Additionally, while consensus scores had significantly different distributions between CD and UC patients, it was not particularly impacted by the metabolite ontology (Supplementary Fig. 3) which reflects the strong relationship occurring between diet and microbial metabolism.

Our study detected several metabolites known to have important roles in the disease state then built connections with host metabolism and disease states. For example, nicotinic acid was connected to its known receptor, GPR109 (HCAR2), an interaction known to reduce inflammation in IBD[24]. This finding was in accordance with experimental evidence showing anti-inflammatory effects of niacin in a HCAR2-dependent manner[25]. Counterintuitively, we found HCAR2 to be significantly over-expressed in CD and UC patients. HCAR2 is often involved in lipid metabolism[26] and colonic inflammation[24]. We also found that trigonelline, another activator of HCAR2, was over-represented. Trigonelline has protective effects in IBD DSS mice models[27], but is also involved in lipid metabolism, suggesting its role in the HCAR2 upregulation[28].

Nicotinuric acid, degradation product of nicotinic acid, was instead connected to LMNA, whose knock-out promotes inflammatory responses in IBD mice models, through increase in CD4$^+$ T-cells[29]. A potency assay built this connection, so the directionality of this interaction with respect to disease is unclear, but might help to explain nicotinuric acid effects on lymphocytes[30]. Histamine was enriched in both UC and CD patients and its cognitive receptor HRH4 was over-expressed. Histamine also induced a pro-inflammatory in vitro profile. Collectively, these findings are well-aligned with the proposed contribution of an activated histamine-HRH4 axis in other inflammatory disorders such as Meniere disease[31].

Long chain polyunsaturated fatty acids (i.e., arachidonic acid, docosapentanoic acid and 8,11,14-eicosatrienoic acid) in IBD are known pro-inflammatory metabolites[32] and we detected their significant enrichment in HMP2 study patients. Other metabolites with high consensus score but yet unknown disease implications belong to compound classes associated with either anti-inflammatory (phenylpropanoids[33]) or pro-inflammatory (cholesteryl esters[34]) activities.

A deeper understanding of the mechanistic interaction in this disease space is still necessary for the development of IBD therapeutics[35]. By leveraging a virtual screening approach, we connected metabolites to compounds with measured modulation properties recorded in ChEMBL using 2d-similarity screening via molecular fingerprints. We limited our searches to functional assays in order to have immediate and confident direction of modulation between the compound and the target protein. Purposely, we did not extend computational screening to QSAR models to avoid known pitfalls of predictive associations[36] and only defined "positive" modulation (which includes agonist, activators, cofactors, etc.),

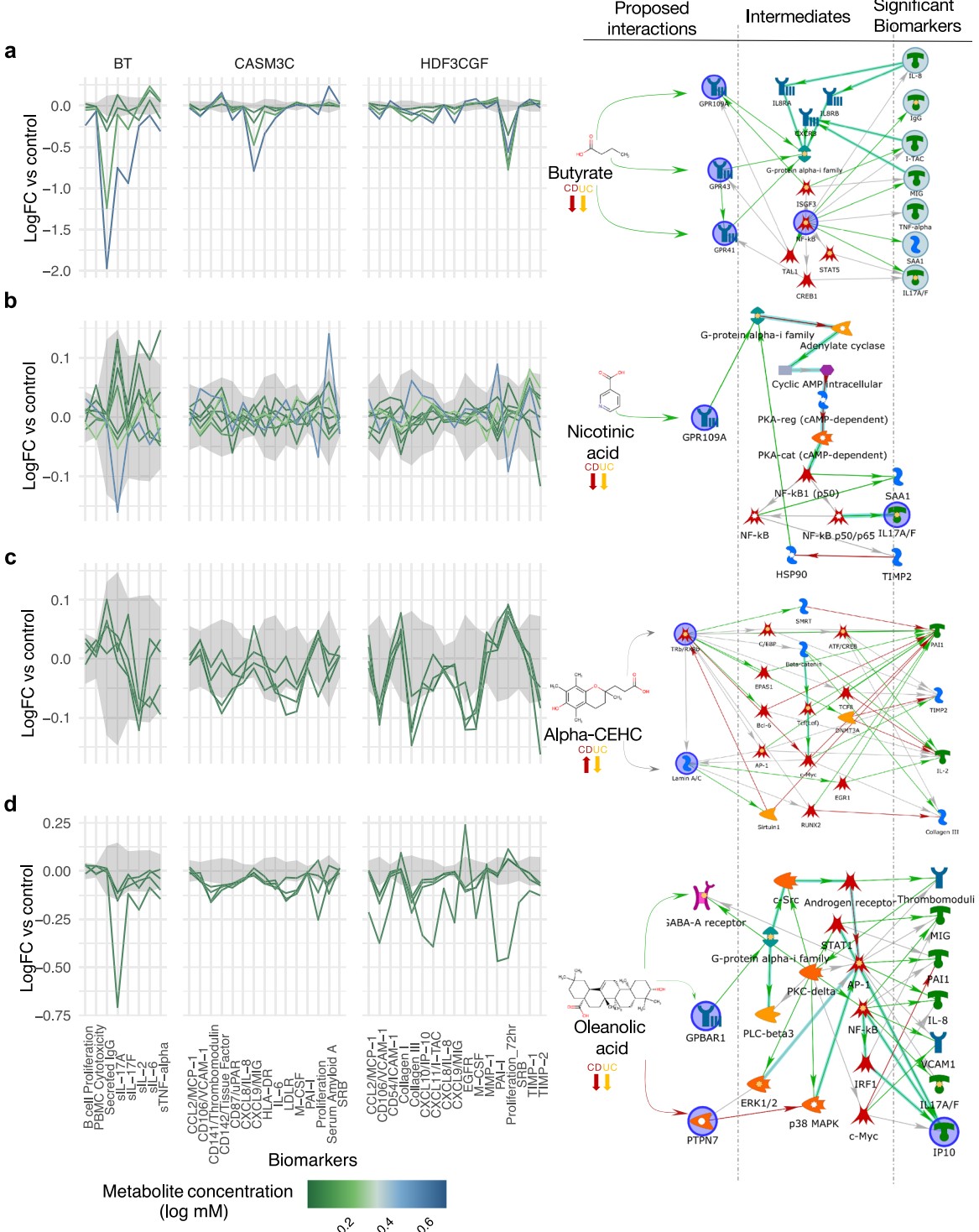

**Fig. 5 Examples of biomarker readouts from in vitro cell assays for four of eleven tested metabolites. a** Butyrate, **b** Nicotinic acid, **c** Alpha-CEHC and **d** Oleanolic acid. Metabolites were administered at different concentrations, here ranked from higher to lower (concentrations in Supplementary Data 8). Readouts graph show the differential abundance vs baseline for the B and T cell system (BT), arterial smooth muscle cells (CASM3C) and wound healing (HDF3CGF) (full results for all 11 metabolites are shown in Supplementary Fig. 6). Knowledge-based graphs on the right represent possible pathway connections between the proposed targets for each metabolite and the most significant biomarker readouts. Interactions are color-coded for positive (green), negative (red) and unknown (gray) modulation.

"negative modulation" (including inhibitors, antagonists, etc.) or "other" for all uncharacterized or unclear interactions. Additionally, we used a target classification based on mechanism of action for drug and treatment[37] and found that the most connected proteins

were well-known drug targets such as GPCRs, transcriptional factors and various enzymes.

Several host–metabolite pairings emerge from our analyses which have yet to be explored for drug purposes to the best of our

knowledge. Hydrocinnamic acid (or phenylpropanoid acid), which is ingested through seeds and metabolized by *Clostridia* species, has known anti-inflammatory properties[33] and was depleted in both UC and CD patients. Hydrocinnamic acid has a mild inhibition effect on toll-like receptor 7 (TLR7) which is an activator of innate immunity NF-κB expressing cells[38], thus might be useful to dampen the inflammatory response.

Oleanolic acid is a plant triterpenoids with anti-inflammatory properties[39]. We connected oleanolic acid (and its analog ursanolic acid) to PTPN7 which interacts with MAP-kinases to lower expression of downstream NF-κB[40]. Oleanolic acid was also connected to GPBAR1, which has anti-inflammatory activity[41], as well as GABRG2, which is genetically associated with an increase in macrophage inflammatory protein 1b. The directionality between the GABRG2 genetic variant and the inflammatory phenotype needs further confirmation. In vitro assays performed in this study, provide further confirmation of the immune-modulatory and anti-inflammatory properties of oleanolic acid and lends support to further determination of mechanism of action, perhaps starting with GPBAR1 and PTPN7 mediated responses.

GABRG2 was also connected to several bile acids conjugates. Imbalances between conjugated bile acids operated by the microbial metabolism, such as the detected increase in glycocholate and taurocholate severely impacts the inflammation mechanism through modulation of the farnesoid receptor (FXR)[42].

GPR119 is an orphan GPCR (i.e., its endogenous ligand has not been yet identified), which was downregulated in the UC cohort. GPR119 has also been previously linked to endocannabinoid metabolites[43] and shows similar mechanistic anti-inflammatory properties in colitis, through release of the glucagon-like peptide GLP-1[44]. Using in vitro assay data, we connected GPR119 with a strong activator underrepresented in IBD patients, linoleoyl ethanolamide, which has been shown to lower LPS-induced macrophage inflammation in dermatitis[45]. We also made another putative connection between L-acetylcarnitine and CHRNB2, both of which were associated separately in neuronal diseases[46].

Retrospective studies suggest that drugs targeting human genes with genetic associations to disease mechanisms might have a higher probability of success in the clinic[47,48]. Therefore, we sought to align target-metabolite pairings with genetic association to IBD- or inflammation-related phenotypes. The genes NOS2, CXCR1 and CXCR2, which have robust genetic associations with such phenotypes, were also connected to CD or UC disease states through metabolomics and transcriptomics. For example, the connection built between depletion in L-arginine and enrichment in NOS2 confirms the antioxidant effects of L-Arg[49]. Serotonin receptor HTR4 is another target involved in both the gut-brain axis and inflammation. We found serotonin depletion to the advantage of 5-HTP, its precursor. 5-HTP enrichment has been previously reported for IBD[50] and gut microbiota have a key role in modulating serotonin synthesis and regulating 5-HTP production[51]. We have also confirmed anti-inflammatory effects of serotonin in vitro cell assay (Supplementary Fig. 6).

CXCR1 and CXCR2 were overexpressed in IBD patients while ibuprofen, a negative modulator, was under-represented. Ibuprofen, a well-known nonsteroidal anti-inflammatory drug (NSAID), was derived from a natural metabolite, propionic acid[52]. NSAIDs possibly promote exacerbation events in IBD patients, however, a recent meta-analysis of published data failed to find a statistically significant association between NSAID usage and colitis occurrence[53]. It is unclear whether this cohort of IBD patients was prevented to assume ibuprofen by medical prescription, but the causal determination of such interactions is beyond the scope of this study.

Interestingly, the connections we built included potential biomarkers of disease severity. For example, 7-methylguanine, a biomarker of colorectal cancer[54], was connected through 8-aminoguanine with purine nucleoside phosphorylase (PNP). PNP deficiency is responsible for T-cell lymphopenia[55]. The anti-directional evidence might suggest that, in the HMP2 cohort, increase in 7-methylguanine does not modulate the immune system, but is rather a consequence of the ongoing dysbiosis in these patients, possibly due to the detected overabundance of *E. coli*[9].

In conclusion, we identified 983 high quality hypothetical connections between gut microbial metabolites and human proteins with potential relevance to IBD and other immune-related diseases. These proposed connections require further experimental validation in order to establish their direct role in disease causality or progression, rather than a consequence of disease dysfunction. Nonetheless, our study highlights the relationship importance of diet and human-hosted microbiota in modulating the immune system responses and provides perspective metabolite–target connections for drug design purposes.

## Methods

**Statistics reproducibility.** Metabolomics and host transcriptomics data were downloaded from the public repository of the HMP2 project, the Inflammatory Bowel Disease Multi-omics Data Base (https://ibdmdb.org/). Samples description, collection, replicates, and preprocessing analyses are described in Lloyd-Price et al.[9]. Of the original 132 participants only those with metabolomics or transcriptomics samples were included in this analysis, resulting in the sample size and distributions described in Table 1.

**Metabolomics data analysis.** Peak areas for each metabolite were normalized for each LC/MS method using total sum scaling and isometric log-transformation. The values were then averaged across methods to result in a table of metabolite by samples. PCA was used to filter outlier samples (only two outlier samples were filtered out).

Bootstrap-coupled estimates[56] were performed on each metabolite singularly (5000 iterations) using multi-control grouped design (i.e., UC vs nonIBD and CD vs nonIBD) and significance was assessed via Mann–Withey U test (p values were FDR-corrected with Benjamini & Hochberg method)[57].

Machine learning analysis was performed as follows: six machine learning classifier methods from Python packages Scikit-learn 0.21.3[58] (logistic regression, k-nearest neighbors, random forest, 3-layers dense neural net, gaussian naïve Bayes, linear kernel one-vs-rest), XGBoost 0.90[59] and a novel generalized version of mixed effects random forest model (with the XGBoost kernel)[60] were tested with default parameters to predict diagnosis labels from the transformed metabolomics data matrix. Recruitment site, use of antibiotics and patient age metadata were included in the training matrix as additional features. A 10-fold stratified cross validation was used to avoid overfitting. All methods performances were assessed with the weighted F1-score of the predictions on the holdout test set (15% of the dataset) and the best method was selected for the highest score (XGBoost).

XGBoost was also trained separately with additional hyperparameter tuning using GridSearch 5-fold cross-validation. Then, a generalized mixed effects machine learning model with an XGBoost core, based on the generalized mixed effects random forest[60], was trained using participant ID as cluster, hospital site as random effect and sex, age and antibiotics as fixed effect. Explanatory metabolite weight per disease state prediction was assessed for XGBoost and mixed effect XGBoost using feature importance, gain and SHAP values[59].

Finally, the following features were scaled in the interval [0,1] and combined to generate the consensus score: (i) FDR-adjusted p value from the Mann–Whitney U test on the bootstrap-coupled estimation; (ii) feature gain per metabolite in the XGBoost model; (iii) feature importance per metabolite in the XGBoost model; (iv) SHAP value per metabolite per disease state in the XGBoost model; (v) feature importance per metabolite in the generalized mixed effects XGBoost model; (vi) SHAP value per metabolite per disease state in the generalized mixed effects XGBoost model. Consensus score was computed by averaging the selected scaled predictors scaling the average between [0,1] and square-root-normalization. Definition of metabolite classes and origin were parsed from The Human Metabolome Database 4.0 (HMDB)[61] and summary visualizations were built using Upset visualizations[62].

**Host transcriptomics data analysis.** Uniform Manifold Approximation and Projection (UMAP)[63] visualizations were built using R package uwot v0.1.8. Count data from biopsies were normalized using R package DESeq2 1.22.2[64]. Due to the important effect of biopsy location on expression data, samples were assigned to a dummy variable representing either ileum or non-ileum biopsies. Differential gene

expression per each disease state was analyzed using a $\sim diagnosis*is\_ileum$ formula with DESeq2. Pathway enrichment analysis was performed using protein interaction networks in R package pathfindR 1.3.0[65] on the Reactome database[66]. Definition of the target classes of interest for drug design purposes were assigned using a pre-defined target map[67].

**Similarity searches and ligand-based virtual screening**. All metabolites originally present from the metabolomics dataset were used as queries for similarity searches (i.e., before any filtering step). HMDB was used to parse SMILES strings for each metabolite (with additional manual curation to fix mismapping/missing HMDB codes from the original HMP2 study). SMILES strings were converted into 2D chemical fingerprint using Python package RDKit. Similarity searches were performed using Tanimoto and Tversky$_{\alpha=0.05}$ similarity scores against compounds in the ChEMBL database v25 (https://www.ebi.ac.uk/chembl)[68] internally ingested, using ChemAxon MadFast Similarity Search software with default parameters. The full hypothesis space included 2,721,397 unique compound–target pairs from the ChEMBL database. Assays data were then parsed for each compound using their unique ChEMBL ID numbers.

**Target selection from genome-wide association studies**. Genetic association to IBD, CD and UC an inflammatory conditions was parsed from the GWAS catalog[69] (release 2020-07-14) and looking for disease traits containing the keywords "inflammatory", "Crohn", "colitis", "monocyte", or "lymphocyte". Mendelian evidence was parsed manually from The Online Mendelian Inheritance in Man database (OMIM)[70] from reference #266600 which includes Crohn's Disease and Ulcerative Colitis. Additional candidate targets were added manually from a recent comprehensive reviews of IBD-related pathways[14]. Networks representations are optimized manually via RCy3 and Cytoscape 3.8[28].

**In vitro validation assays**. In vitro assays were performed using the Discovery PLUS platform of the BioMAP panel[21] at Eurofins Panlabs, Inc. (St Charles, MO, USA). BioMAP systems are constructed with one or more primary cell types from healthy human donors, with stimuli (such as cytokines or growth factors) added to capture relevant signaling networks that naturally occur in human tissue or pathological conditions. Conditions tested are as follows: vascular biology model for inflammatory environment Th1-specific (*3C*) and a Th2-specific (*4H*); Th1 inflammatory state specific to arterial smooth muscle cells (*CASM3C*); monocyte-driven Th1 inflammation (*LPS*); T cell stimulation (*SAg*); chronic Th1 inflammation driven by macrophage activation (l*Mphg*); T cell-dependent activation of B cells that occurs in germinal centers (*BT*); Th1-specific (*BE3C*) and Th2-specific (*BF4T*) airway inflammation of the lung; myofibroblast-lung tissue remodeling (*MyoF*); skin biology (*KF3CT*) and wound healing (*HDF3CGF*). Protein biomarker readouts are selected for predictiveness of their mechanism of action. Compounds concentrations were selected based on compound solubility and minimizing cytotoxicity (Table 1). After analysis the most important readouts are assigned to the perspective targets through selected shortest paths on the canonical pathway analysis performed via MetaCore (GeneGo) v20.1 (Thomson Reuters, https://portal.genego.com/).

**Reporting summary**. Further information on research design is available in the Nature Research Reporting Summary linked to this article.

## Data availability
All data from the IBD cohort are available at the IBDMDB website (https://ibdmdb.org), including cohort description and sample handling and preprocessing. All pertinent database are publicly available: HMDB[61], ChEMBL[68], OMIM[70] and GWAS catalog[13]. All other data are provided in Supplementary Data files. Any remaining information is available from the corresponding author upon reasonable request.

## Code availability
Scripts to reproduce the analyses are available as downloadable repository[71] under Creative Commons 4.0 open license; including instructions to build a Docker image for reproducibility (https://doi.org/10.5281/zenodo.4439416).

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

## Acknowledgements

The study was funded through GSK Consumer Health through a postdoctoral fellowship. The authors would like to David Cooper, Valeriia Sherina, Hoang Tran, Pat Brady, Dinesh Manahdar and Qing Xie at GSK Pharma R&D for their support and technical advice. We would like to also acknowledge the HMP2 study authors for making their data and computer code publicly available.

## Author contributions

A.N., J.T. and J.R.B. conceived the study. A.N. analyzed the data. J.T. E.B., S.S. and C.J. performed initial pilot studies and BioMAP profiling. A.N. and J.R.B. wrote the manuscript. All authors reviewed and revised subsequent manuscript drafts.

## Competing interests

A.N., S.S., C.J., J.T and J.R.B. were all employees of GlaxoSmithKline at the time of this study. E.B. is an employee of Eurofins.
