## [Peer Review File · Communications Biology]

Reviewers' comments:

Reviewer #1 (Remarks to the Author):

The authors propose a very novel approach to expand on drug space by proposing specific associations between metabolite and receptors. Similar concepts have been proposed previously (EMBO Mol Med. 2020 Apr 7;12(4):e11621). The work here is novel and new because new targets of this concept are being proposed. That is they argue that by defining metabolite -host receptor pairings, the metabolite could then become a launch point for the discovery of new analogs as drugs based on the metabolite itself. To do this, they employed computational approaches to identify 983 metabolite-receptor pairings and then validated 4 of these using in vitro based confirmatory ligand-receptor assays (BioMAP panel). The original data mining set came from the already published HMP2 project. The overall strength of the techniques used, computationally speaking seems strong and supports their association claims. The limited experimental determination to 4 such pairings is underwhelming but the authors do provide us with the window to perform more such assays. This alone is not a major issue. Some aspects that the authors should provide some detail on are the following:

1. In the HMP2 biopsies and assays on RNA targets (e.g., microarrays etc..) were performed but it is unclear as to when the fecal metabolites were assessed in relation to this? Were the feces parallel processed at the same time the biopsies were taken ? In other words, there needs to be some detail on the timeline of acquisition of samples. Furthermore, a single sample is fraught with error for metabolomics - if this was the case then the authors should describe pitfalls of their assessments and analysis in more depth in the discussion.

2. It is unclear why only 4 metabolites were tested in functional assays ? In other words, are all the hypothetical pairings in fact true - or are there some negative ones - it is important to describe negative assays as it builds into the reliability of the computational work.

Overall, this is an important theoretical concept expanding a new area of drug discovery and biology and so worthy of general readership.

Reviewer #2 (Remarks to the Author):

Nuzzo et al. combine metabolome and transcriptome data from the HMP2-IBD dataset with compound activity data and GWAS data. In this interesting combination of publicly available data, they propose novel candidates for interactions between gut metabolites and differentially expressed genes that could serve as potential drug targets.

The individual steps taken seem sensible to me. In Fig 1f, the authors show that there is a very strong correlation between their newly developed score and the HMP2 scores. How do the new methods outperform the HMP2 method?

There are a number of assumptions that are not stated explicitly, and which need to be made clear and backed up by evidence from the literature. For example: "Differentially abundant compounds are good drug candidates." "Differentially expressed genes are good drug targets." (And in general: "Differentially abundant compounds and genes are contributing causally to the diseases, as opposed

to being downstream effects.”) “Genes detected in GWAS are good drug targets.”

If these assumptions can be made clear, this paper and its data can become a useful resource for scientists investigating potential treatments for IBD.

More attention is needed to details:

The manuscript is about microbial metabolites, but ibuprofen is presented as an example compound (line 157). How can this be? Since this is a drug, is there any evidence of this being useful for the treatment of IBD?

Line 82: “five different machine learning methods” while the methods mention six methods

Line 94: LogFC – is this natural logarithm? Log 10 / 2?

Line 112 and many others: What does “pXC50” mean? I guess this is the author’s non-standard way of denoting pIC50 / pEC50, but not really clear.

Table 2: The connections between the gene and ligand columns is not clear when cells have been merged in the left column.

Fig 1 and others: please ensure that colors are chosen so that people with red-green color blindness can still distinguish them.

Fig 1b: Colored text in front of colored dots is not legible.

Fig 1cde: Please move the panels together like in the traditional Upset plot (especially remove the gap between c and e), otherwise it’s not immediately clear that e is connected to c.

Fig 2: The order of the legend and of the target class in the plot is reversed, making it harder to read the plot.

Fig 5: Text on the right is too small to be legible.

Clean up Supplementary tables: Rows are hidden, apparently random cells selected

Reviewer #3 (Remarks to the Author):

This is an interesting paper that investigate the potential of screening drugs based on metabolites of patients. My comments can be found below.

(1) The authors present several different ways to prioritize metabolite-protein pairs. How to combine these together?

(2) I'm wondering what is the relationship between the compounds they found with those drugs

used in clinical? Are they chemically similar?

(3) It seems something missing in the paper, which is also highlighted already in the manuscript.

Nuzzo et al Manuscript COMMSBIO-20-2547-T: Detailed response to Reviewers' comments

No.	Reviewer Comments	Our Response
	Reviewer #1	
	The authors propose a very novel approach to expand on drug space by proposing specific associations between metabolite and receptors. Similar concepts have been proposed previously (EMBO Mol Med. 2020 Apr 7;12(4):e11621). The work here is novel and new because new targets of this concept are being proposed. That is they argue that by defining metabolite-host receptor pairings, the metabolite could then become a launch point for the discovery of new analogs as drugs based on the metabolite itself. To do this, they employed computational approaches to identify 983 metabolite-receptor pairings and then validated 4 of these using in vitro based confirmatory ligand-receptor assays (BioMAP panel). The original data mining set came from the already published HMP2 project. The overall strength of the techniques used, computationally speaking seems strong and supports their association claims. The limited experimental determination to 4 such pairings is underwhelming but the authors do provide us with the window to perform more such assays. This alone is not a major issue. Some aspects that the authors should provide some detail on are the following:	We thank the reviewer for their interest and overall support of our study.
1	In the HMP2 biopsies and assays on RNA targets (e.g., microarrays etc.) were performed but it is unclear as to when the fecal metabolites were assessed in relation to this? Were the feces parallel processed at the same time the biopsies were taken? In other words, there needs to be some detail on the timeline of acquisition of samples. Furthermore, a single sample is fraught with error for metabolomics - if this was the case then the authors should describe pitfalls of their assessments and analysis in more depth in the discussion.	All sample timelines, sampling protocols and frequencies are described in the original IBD HMP2 paper of Lloyd-Price et al. 2019 (https://doi.org/10.1038/s41586-019-1237-9) and shown in their Figure 1c,b. Their multi-omic measurements were either from the same sample (strict) or near-concordant time points (with differences of up to 2 or 4 weeks). Lines 75-77. Here we state that for our analysis, "Patients with less than 3 samples per datatype or with only one sampling point were excluded, to the final sample size described in Table 1".

		With respect to the metabolomics replicates, we used the preprocessed data from the HMP2 project (we did not perform the metabolomics analysis on the samples ourselves). The full procedure is described on the IBDMDB portal (https://www.ibdmdb.org/cb/document/Data%20Generation%20Protocols/MetabolomicsHMP2Protocol.pdf) but briefly, every stool samples was aliquoted into 4 different aliquots, one per each LC/MS method. In addition, pooled samples were used as reference for temporal drifts. Some samples were replicated by the HMP2 team for not passing the QC analysis and we discarded the low-quality samples and the outliers from our analysis. This was reported in the Supplementary Methods, lines 27-30.
2	It is unclear why only 4 metabolites were tested in functional assays? In other words, are all the hypothetical pairings in fact true - or are there some negative ones - it is important to describe negative assays as it builds into the reliability of the computational work.	We included in vitro functional assays results for a further 7 metabolites: 13-cis-retinoic acid, acetylcholine, adenosine, histamine, ibuprofen, lithocholic acid, and serotonin. Supplementary Table 7 and Supplementary Fig. 6. Updated with additional data Line 34. Abstract amended to read 11 metabolites. Lines 207-242. Results subsection modified and restructured to include these new results as well as retain the focus on 4 metabolites-targets with interaction maps shown in Figure 5. Lines 277-280 added. “Histamine was enriched in both UC and CD patients and its cognitive receptor HRH4 was over-expressed. Histamine also induced a pro-inflammatory in vitro profile. Collectively, these findings are well-aligned with the proposed contribution of an activated histamine-HRH4 axis in other inflammatory disorders such as Meniere disease³². To the reviewer’s comment about describing negative assays, we have already done so with respect to the low activity of Nicotinic acid and Alpha-CEHC See Lines 230-235: “Nicotinic acid (vitamin B3) is an anti-inflammatory activator of HCAR2. Nicotinic acid was largely inactive at tested concentrations with minor lowering of soluble IL-17A in the BT system (Fig. 5b). Alpha-CEHC was a highly scoring metabolite in our analysis

		with unclear directionality to disease mechanisms. Alpha-CEHC also had low activity with slight suppression of several inflammatory markers in the HDF3CGF system of dermal fibroblast cells modeling wound healing and fibrosis (Fig. 5c)."
	Reviewer #2	
	Nuzzo et al. combine metabolome and transcriptome data from the HMP2-IBD dataset with compound activity data and GWAS data. In this interesting combination of publicly available data, they propose novel candidates for interactions between gut metabolites and differentially expressed genes that could serve as potential drug targets.	We thank the reviewer for their interest and overall support of our study.
1	The individual steps taken seem sensible to me. In Fig 1f, the authors show that there is a very strong correlation between their newly developed score and the HMP2 scores. How do the new methods outperform the HMP2 method?	The two methods serve different purposes: while the HMP2 team was essentially looking for metabolites associated with the disease (i.e. markers), our analysis focuses on mechanism of action and generating hypothesis for host receptor binding/manipulation (i.e modulators). Therefore, while reasonable to expect concordance between the two methods, it is not possible to establish a common metric that could compare the performances of the two. The authors of the HMP2 study applied linear models to determine metabolites differential concentrations in healthy vs disease with an additional covariate, the dysbiosis score, computed on the beta-diversity of the microbial communities of the stool samples. Being based on beta-diversity analyses, the dysbiosis score is a non-reproducible across multiple cohorts in itself and poses problems for application to other studies. We also attempted to reproduce the HMP2 analysis without the dysbiosis score through the published code but it yielded no significant results (not shown in our manuscript for conciseness). Our consensus scoring method does not recur to a dysbiosis score in itself, proving to be more flexible and generalizable to cross-cohort studies, which is an improvement for drug discovery purposes compared to the HMP2 study. We previously commented on issues with dysbiosis scoring reproducibility on lines 249-252: "Compared to the HMP2 IBD study, our

		consensus scoring retrieved more relevant metabolites without recurring to a dysbiosis score, an index derived from the beta-diversity analyses of the metagenomic specimens, which poses issues for reproducibility across cohorts and translatability to treatment purposes²².”
2	There are a number of assumptions that are not stated explicitly, and which need to be made clear and backed up by evidence from the literature. For example: “Differentially abundant compounds are good drug candidates.” “Differentially expressed genes are good drug targets.” (And in general: “Differentially abundant compounds and genes are contributing causally to the diseases, as opposed to being downstream effects.”) “Genes detected in GWAS are good drug targets.” If these assumptions can be made clear, this paper and its data can become a useful resource for scientists investigating potential treatments for IBD.	As clarification, we do not explicitly state that differentially abundant compounds are good drug candidates nor that differentially expressed genes are good drug candidates. Rather genes or metabolites which have differentially abundances in IBD vs non-IBD patients might lead to a new understanding of disease mechanisms. In the Introduction, we clearly state our objective which was to expand the universe of hypothetical metabolite-host cross-talk since these potential interactions might be therapeutically modulated for IBD as well as other immune-inflammation diseases (see lines 42-52 and references therein). We also presented several examples where metabolite and gene expression patterns were in accordance with experimental evidence for IBD, such as increased abundance of trigonelline, a known activator of HCAR2 that was also upregulated (lines 163-166 and 262-271). We also show several examples of co-directionality between metabolite and gene transcript abundances (lines 168-176). In response to the specific comment, “Differentially abundant compounds and genes are contributing causally to the diseases, as opposed to being downstream effects.” Lines 362-364 were added: “These proposed connections require further experimental validation in order to establish their direct role in disease causality or progression, rather than a consequence of disease dysfunction.” With respect to the comment that “Genes detected in GWAS are good drug targets”, we made the following changes: Lines 332-333 amended: “Retrospective studies suggest that drugs targeting human genes with

		genetic associations to disease mechanisms might have a higher probability of success in the clinic^{45,46}. Therefore, we sought to align target-metabolite pairings with genetic association to IBD- or inflammation-related phenotypes.”
3	The manuscript is about microbial metabolites, but ibuprofen is presented as an example compound (line 157). How can this be? Since this is a drug, is there any evidence of this being useful for the treatment of IBD?	Ibuprofen was present in the original HMP2 dataset (originally named “carboxyibuprofen”, one of its synonyms), with a non-null peak before normalization in 179/536 samples (~33%). Ibuprofen and other NSAIDs have been associated with IBD exacerbations in multiple clinical case studies. However, a recent meta-analysis of published data did not find a statistically significant association. Lines 156-161 were amended: “Several metabolites underrepresented in IBD were classified as tentative negative modulators of upregulated targets. For example, receptors of the CXC ligand 8 (CXCL8 or IL8) chemokine family, CXCR1 and CXCR2, were overexpressed while their known negative modulator compound, ibuprofen (pXC50 = 7.0) and its HMP-2 derivative, 2-hydroxyibuprofen, (Supplementary Table 3), were underrepresented in IBD patients although below the consensus scoring threshold (Supplementary Table 1).” Lines 337-341 (Discussion) added: “CXCR1 and CXCR2 were overexpressed in IBD patients while ibuprofen, a negative modulator, was underrepresented. Ibuprofen, a well-known nonsteroidal anti-inflammatory drug (NSAID), was derived from a natural metabolite, propionic acid⁵³. NSAIDs possibly promote exacerbation events in IBD patients, however, a recent meta-analysis of published data failed to find a statistically significant association between NSAID usage and colitis occurrence⁵⁴. It is unclear whether this cohort of IBD patients was prevented to assume ibuprofen by medical prescription, but the causal determination of such interactions is beyond the scope of this study.” We also respect the reviewer’s comment that our analysis identified putative human host

		targets for metabolites generated from microbial as well as human metabolic processes. Therefore, we amended sentences in the abstract and introduction. Line 24 amended: “Metabolites produced in the human gut are known modulators of host immunity.” Line 42-43 amended: “Endogenous metabolites produced in the gastro-intestinal tract (GIT) by microbial and human metabolic processes have a significant role in modulating host immune responses¹.”
4	Line 82: “five different machine learning methods” while the methods mention six methods	We appreciate the two were confusing. Six machine learning methods were tested, and of those six, only one was selected for downstream analysis. From the downstream analysis, feature importance and SHAP values were extracted from both a vanilla version and a generalized-mixed effect flavor of the selected machine learning model. This sums up with a total of 2x2 features plus the estimated power analysis from the Mann-Withey algorithm (which is technically not a machine learning algorithm), to a total of 5 features. These 5 features were combined in a single score. Lines 82-87 amended: “To better define metabolite relevance for CD or UC etiology we utilized an ensemble method that combined results from multiple analytical methods (specifically, power estimation and both feature importance and SHAP values, each from two selected machine learning methods) combined into a single consensus score, normalized between 0 and 1, where 1 represents the most significant metabolite across all methods (Supplementary Fig. 1 and 2; Supplementary Table 1).” Supplementary Material File Lines S55-56 were amended: “Finally, the following features were scaled in the interval [0,1] and combined to generate the consensus score.”
5	Line 94: LogFC – is this natural logarithm? Log 10 / 2?	We have specified for each occurrence whether it was base 2 or base 10 logarithm.
6	Line 112 and many others: What does “pxC50” mean? I guess this is the author’s non-standard way of denoting pIC50 / pEC50, but not really clear.	Line 111-115 were amended: “We further filtered for compounds having high similarity scores with the top-ranking metabolites (i.e., Tanimoto similarity ≥ 0.85 or Tversky $\alpha=0.05$ similarity ≥ 0.95) and, for those, only binding

		proteins with perspective high affinity (i.e., either pIC50 or pEC50 values, or pxC50 \geq 5.5) were retained.”
7	Table 2: The connections between the gene and ligand columns is not clear when cells have been merged in the left column.	We have amended the formatting of the table and we will be conscious of the issue during the editorial process.
8	Fig 1 and others: please ensure that colors are chosen so that people with red-green color blindness can still distinguish them. Fig 1b: Colored text in front of colored dots is not legible. Fig 1cde: Please move the panels together like in the traditional Upset plot (especially remove the gap between c and e), otherwise it's not immediately clear that e is connected to c. Fig 2: The order of the legend and of the target class in the plot is reversed, making it harder to read the plot. Fig 5: Text on the right is too small to be legible. Clean up Supplementary tables: Rows are hidden, apparently random cells selected.	We have updated figures and tables. Colors were also checked by using the Color Oracle software (http://www.colororacle.org/).
	Reviewer #3	
	This is an interesting paper that investigate the potential of screening drugs based on metabolites of patients.	We also thank the reviewer for their interest and overall support of our study.
1	The authors present several different ways to prioritize metabolite-protein pairs. How to combine these together?	We did not combine methods to prioritize metabolite-protein pairs since the approaches are distinct and complementary. Potentially those metabolite-protein pairs showing concordant positive or negative gene expression with metabolite abundances might be more highly ranked as potential drug targets. Similarly, targets with human genetic evidence to IBD might be also be of interest.
2	I'm wondering what is the relationship between the compounds they found with those drugs used in clinical? Are they chemically similar?	A comprehensive comparison of known drugs to natural metabolites is beyond the scope of our study and has been reported elsewhere (i.e. Dobson et al. 2009. 'Metabolite-likeness' as a criterion in the design and selection of pharmaceutical drug libraries. Drug Discovery Today 14:31 https://doi.org/10.1016/j.drudis.2008.10.011). The standard of care for IBD includes anti-inflammatory drugs such as corticosteroids and aminosalicic acids (5-ASA) as well as biologics

		(i.e. monoclonal antibodies) which are not similar to known metabolites. Dietary and surgical intervention are also used (for a review see http://dx.doi.org/10.1136/gutjnl-2019-318484). There is a high unmet medical need for more efficacious treatments for both CU and CD, hence the motivation for our study.
3	It seems something missing in the paper, which is also highlighted already in the manuscript.	Comment is unclear and, after consultation with the Editor, does not require a response.

REVIEWERS' COMMENTS:

Reviewer #1 (Remarks to the Author):

The revisions have adequately addressed my previous concerns. The inclusion of additional metabolite pairings is convincing and adds bidirectionality to the conclusions. I have no new concerns.

Reviewer #2 (Remarks to the Author):

The authors have addressed my concerns and I recommend the publication of this manuscript.